# Strategic Framework for Natural Disaster Risk Mitigation Using Deep Learning and Cost-Benefit Analysis

Ji-Myong Kim[1], Sang-Guk Yum[2], Hyunsoung Park[3], and Junseo Bae[4]

[1]Department of Architectural Engineering, Mokpo National University, Mokpo 58554, South Korea; jimy6180@gmail.com
[2]Department of Civil Engineering, Gangneung-Wonju National University, Gangneung 25457, South Korea; skyeom0401@gwnu.ac.kr
[3]Department of Mechanical and Civil Engineering, University of Evansville, Evansville, Indiana 47722, United States; hp93@evansville.edu
[4]School of Computing, Engineering and Physical Sciences, University of the West of Scotland, Paisley PA1 2BE, United Kingdom; junseo.bae@uws.ac.uk

*Correspondence to*: Junseo Bae (junseo.bae@uws.ac.uk)

**Abstract.** Given trends in more frequent and severe natural disaster events, developing effective risk mitigation strategies is crucial to reduce negative economic impacts, due to the limited budget for rehabilitation. To address this need, this study aims to develop a strategic framework for natural disaster risk mitigation, highlighting two different strategic implementation processes (SIPs). SIP-1 is intended to improve the predictability of natural disaster-triggered financial losses using deep learning. To demonstrate SIP-1, SIP-1 explores deep neural networks (DNNs) that learn storm and flood insurance loss ratios associated with selected major indicators and then develops an optimal DNN model. SIP-2 underlines the risk mitigation strategy at the project level, by adopting a cost-benefit analysis method that quantifies the cost effectiveness of disaster prevention projects. In SIP-2, a case study of disaster risk reservoir projects in South Korea was adopted. The validated result of SIP-1 confirmed that the predictability of the developed DNN is more accurate and reliable than a traditional parametric model, while SIP-2 revealed that maintenance projects are economically more beneficial in the long-term as the loss amount becomes smaller after 8 years, coupled with the investment in the projects. The proposed framework is unique as it provides a combinational approach to mitigating economic damages caused by natural disasters at both financial loss and project levels. This study is its first kind and will help practitioners quantify the loss from natural disasters, while allowing them to evaluate the cost effectiveness of risk reduction projects through a holistic approach.

**Keywords.** Natural disaster; risk mitigation strategy; economic damage; deep learning; cost-benefit analysis

## 1 Introduction

Over the past decades, the frequency and severity of extreme weather events are rapidly increasing due to climate changes. These events represented by flooding, drought, heavy rain, tropical cyclone, heat waves or cold waves have often caused various damages in not only short term, but also various long-term effects such as sea level rises and disease spreads. The negative impact of these event has been warned by the Intergovernmental Panel on Climate Change (The Fifth Assessment Report, 2014). Nevertheless, across the world, severe weather events such as typhoons, heavy rains and changing patterns of meteorological disasters have already increased the loss of many lives and built assets. These damages are still expected to be accelerated in coming future (Kim et al., 2020).

Given the continuous trend, it is well known that natural disaster-triggered losses have been very closely tied with many economic losses worldwide. For example, Western European countries such as France, Germany, and Switzerland were hit by three consecutive tropical cyclones (e.g., Anatol, Lothar, and Martin) in 1999, resulting in a loss of 13 billion euros (Ulbrich et al., 1999). Typhoon Haiyan, which hit the Philippines and China of South Asia in 2013, was one of Category 5 Super Typhoons, was the most extreme tropical cyclone recorded on land. The typhoon's life-threatening wind and rain were enough to smash properties. South Asian countries adjacent to the typhoon track inflicted about $300 billion in damage (Kim et al., 2019). Hurricane Katrina that hit the South Eastern areas in United States in 2005 caused the most severe damage in the national historic record as a Category 5 tropical cyclone. In detail, it caused the US Gulf Coast city to have $180 billion in direct and indirect damages due to substantial rain and robust winds (Blake et al., 2007). Later, in 2017 solely, three different strong hurricanes named by Harvey, Maria, and Irma caused together a total damage amount of about $293 billion, based on the individual damage amounts of $125 billion by Harvey, $90 billion by Maria, and $77.6 billion by Irma (USNHC, 2018).

In this sense, the quality of living in the built environment has been threatened by natural disasters in the globe. To reduce these threats, many of non-governmental organizations and countries have investigated in prevention or post-disaster recovery strategies, on aspects of time, budget, and manpower to mitigate natural disaster risks. Mitigation of risks can reduce the loss by decreasing vulnerability or by decreasing the frequency and severity of causal factors (Rose et al., 2007). For risk mitigation, the execution and allocation of financial resources should be carried out promptly and extensively, against the limited resources available. Hence, it is important to estimate strategically the cost impact of natural disaster risks and the effect of risk reduction at the same time, specifically aiming at achieving the ultimate reduction and mitigation of risks through an efficient use of the limited resources.

## 2 Point of Departure: The need of more effective strategic framework for natural disaster risk mitigation

### 2.1 Decision-support for natural disaster risk mitigation strategies

Given the increasing frequency and severity of natural disasters, the demand for sophisticated natural disaster loss forecasting also increases. In response to such demand, various companies and national organizations have developed models to predict natural disaster losses. The New Multi-Hazards and Multi-Risk Assessment Method for Europe (MATRIX) in Europe, the HAZUS-Multi Hazard (HAZUS-MH) by the Federal Emergency Management Agency (FEMA) in the United States, the RiskScape in New Zealand, and the Probabilistic Risk Assessment initiative in Central America are representative models (Kim et al., 2017). Florida, USA, has developed a Florida Public Hurricane Loss Model (FPHLM) to predict losses due to hurricanes as it is located on the main north-facing road of hurricanes (Kim et al., 2020). These models are being used in different regions to assess the loss of life and potential economic losses for buildings and infrastructure owing to natural disasters. Nevertheless, since these models were developed based on the vulnerability of natural disasters and the severity and frequency of natural disasters in specific areas, they could not be applied to other areas.

Companies specializing in natural disaster risk modeling have also developed different models, including EQECAT, Applied Insurance Research (AIR), and RMS (Risk Management Solution) (Kunreuther et al., 2004; Sanders, 2002). These models are widely used by insurers and reinsurers around the world to assess the risk of economic loss from natural disasters (e.g., windstorms, earthquakes, floods, winter storms, and tornadoes). Nonetheless, these models have annual fees that are expensive to small and medium-sized users. In addition, these models are available only for the limited number of major countries (Europe, USA, Japan, China, etc.). Furthermore, it is difficult to optimize them for users since they have difficulties to reflect a user's portfolio, capital, business preference, and so on (Kim et al., 2019).

To reflect characteristics and vulnerabilities of each country associated with various situations of users, it is crucial to evaluate the loss through its own model. In order to develop a loss evaluation model, the development of an in-house model using a deep learning algorithm can be a solution. Recently, the 4th revolution technology (e.g., unmanned transportation, big data, artificial intelligence, IoT, robots, etc.) has been applied to various fields and its effectiveness has been recognized (Gledson and Greenwood, 2017; IPA, 2017). To effectively and efficiently analyze the complexity of various sensors-driven big data, the demand for deep learning applications has been increased dramatically. Given the increasing demand, many research efforts on applying deep learning techniques for risk assessment were made recently (Al Najar et al. 2021; Khosravi et al. 2020; Kim et al. 2021; Moishin et al. 2021; Shane Crawford et al. 2020; Sugiyarto and Rasjava 2020; Yi et al. 2020; Zhang et al. 2022). Especially, for improved natural disaster risk assessment and mitigation, neural networks have been widely used for deep learning in various ways (Khosravi et al. 2020; Moishin et al. 2021; Shane Crawford et al. 2020; Yi et al. 2020). Some researchers developed deep learning models to predict flood events (Khosravi et al. 2020; Moishin et al. 2021). Khosravi et al. (2020) developed a flood susceptibility map using convolutional neural networks (CNN). More specifically, 769 historical

flood locations in Iran were trained and tested based on amounts of soil moisture, slopes, curvatures, altitudes, rainfalls,
geology, land use and vegetation, distances from roads and rivers. In addition, a hybrid deep learning algorithm integrating the
merits of CNN and long short-term memory (LSTM) networks was built to manage flood risks by predicting future flood
events, by training and testing daily rainfall data obtained from 11 sites in Fiji between 1990 and 2019 (Moishin et al. 2021).

Other previous studies focused on post-disaster detection caused by landslides or tornados, which uses remote sensed data
collected from satellites for deep learning (Al Najar et al. 2021; Shane Crawford et al. 2020; Yi et al. 2020). Shane Crawford
et al. (2020) adopted CNN to classify damages of 15,945 buildings affected by the 2011 Tuscaloosa tornado in Alabama. To
this end, the authors used satellited-driven images of trees as the damage classification indicator to estimate wind speeds. In
addition, satellite images were embraced into the CNN-driven deep learning process to detect earthquake-induced landslides
in China (Yi et al. 2020). More recently, Al Najar et al. (2021) estimated accurately ocean depths simulating remote sensed
images using a deep learning technique, which overcomes drawbacks of traditional bathymetry measurement activities to track
the physical evolution of coastal areas against any potential natural disasters or extreme storm events. Previous studies
reviewed reveal consistently that deep learning techniques can overcome shortcomings of existing methods and thus to provide
more accurate and reliable decision-support models for risk assessment and risk-informed mitigation strategies.

In addition to applications of deep learning for location detection or event prediction-focused, as stated earlier, it is important
to quantify negative economic impacts caused by natural disasters. Given the importance of economic damage aspects, Kim
et al. (2021) applied a deep learning technique as a cost-effective and risk-informed facilities management solution. In detail,
the authors generalized maintenance and repair costs of educational facilities in Canada, using deep neural networks that learn
sets of maintenance and repair records, asset values, natural hazards such as tornados, lightening, hails, floods, and storms. In
this sense, this study proposed a deep learning modeling framework to predict financial losses caused by natural disasters.

**2.2 Investment strategies for natural disaster risk mitigation**
Mitigating the risk with efficient investment and operation of resources is a challenging task because risk reduction should be
made in a timely manner, with the limited financial resources. To address these issues, cost-benefit analysis has been widely
adopted (FEMA, 2005; Rose et al., 2007). For instance, efficient use of public resources is indicated when total estimated
profits of a risk mitigation activity surpass the entire cost or are parallel to earnings on investment of both private and public.

Disaster risk mitigation represents mitigating social, environmental, and economic damage caused by natural disasters. Since
economic losses due to natural disasters are hard to minimize or avoid separately, there is an increasing public demand for risk
reduction investments to reduce these economic losses (Bouwer et al., 2007; Shreve and Kelman, 2014). Since resources for
risk mitigation investment are restricted, it is critical to estimate economic costs and benefits in order to determine the
effectiveness and appropriateness of the investment. For instance, the Federal Emergency Management Agency of the United
States has reported that the average cost-benefit ratio is 4 for risk mitigation investment (e.g., structural defence measures
against floods and typhoons, building renovations in preparation for earthquakes, etc.) after reviewing 4,000 natural disaster
risk reduction programs in the United States (Kunreuther et al., 2012; Rose et al., 2007). In addition, studies in developing
countries have shown a high cost-benefit ratio in a study of 21 investment activities such as re-establishment of schools and
forestry in preparation for tsunami (Bouwer et al., 2014).

Despite these high potential benefits, investment in risk reduction for residents living in areas at risk of natural disasters is
restricted (Bouwer et al., 2014). According to Hochrainer-Stigler et al. (2010), since natural disaster risk reduction measures
are focused on short-term outcomes, only about 10% of residents in areas vulnerable to natural disasters receive natural disaster
risk reduction measures in the United States. In the case of a natural disaster risk reduction project, a large initial investment
is required, which reduces the expected profit if performance indicators need to be met in a short period of time. As a result,
policy makers and politicians are reluctant to make bold investments in natural disaster risk reduction. They prefer to provide
economic support after disasters (Cavallo et al., 2013). This phenomenon is also reflected in the budget distribution of disaster
management funds of donations and development agencies. Most (98%) of the budget is allocated to reconstruction or relief.
Only the remaining budget (2%) is allocated to risk reduction (Mechler, 2005). As such, while the need for pre-disaster risk
reduction through proactive disaster investment is widely recognized, the economic impact of natural disaster risk reduction
is often not fully considered in decision-making. Moreover, although cost-benefit analysis is the main decision-making tool
commonly used in investment and financial evaluations by public sectors, natural disaster risk is not sufficiently applied in the
cost-benefit analysis (Hochrainer-Stigler et al., 2010). Natural disasters in public sectors' investment projects were often
overlooked or not evaluated based on the cost-to-benefit comparison (Kreimer et al., 2003). In turn, this study explored natural
disaster risk reduction projects and analyzed the cost effectiveness of the projects adopting a cost-benefit analysis method.
**3 Research objectives and methods**
Given trends in more frequent and severe natural disaster events, developing effective natural disaster risk mitigation strategies
is crucial to reduce negative economic impacts on built assets, due to the limited budget for rehabilitation. To address this need,
this study aims to develop a strategic framework for natural disaster risk mitigation, highlighting two different strategic
implementation processes (SIPs), as depicted in Figure 1.

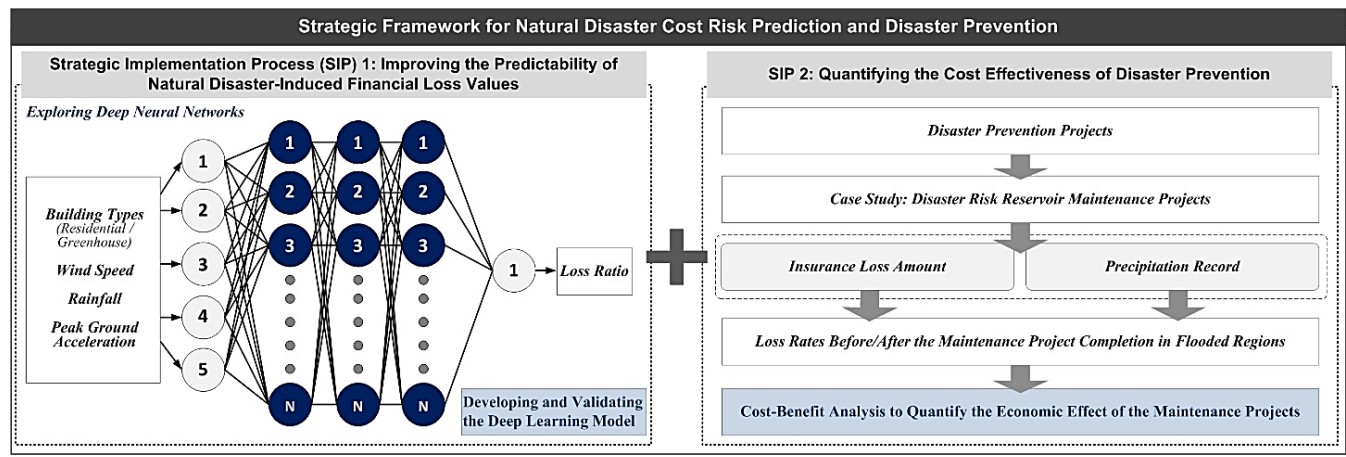

Figure 1. Research framework

More specifically, SIP-1 is intended to improve the predictability of natural disaster-triggered financial loss model. To this end, SIP-1 develops a deep neural network (DNN) model that learns insurance loss amounts to generalize loss ratios, associated with major indicators including rainfall, wind, and ground acceleration. To demonstrate SIP-1, this study collected reliable storm and flood damage insurance data and natural disaster risk indicators, created a predictive model using deep learning, and validate the improved predictability of the model, through the following steps:

1) To collect data on loss caused by natural disasters, this study collected data on claim payout for storm and flood damage insurance from the Korea Insurance Development Institute (KIDI) over the past11 years between 2009 and 2019.

2) This study obtained natural disaster risk indicators based on the collected data.

3) A model of deep learning algorithm was developed using Python 3.7, Keras, and Scikit-Learn libraries. The model was trained, tested, and validated using the collected data.

4) A multiple regression model was independently developed using IBM Statistical Package for the Social Sciences (SPSS) version 23 for model validation.

5) The root mean squared error and mean absolute error values of the deep learning algorithm model and the multiple regression analysis model were estimated and paralleled, respectively.

Compared to SIP-1, SIP-2 underlines the risk mitigation strategy at the project level, by proposing a methodological implementation process for quantifying the cost effectiveness of natural disaster risk reduction by adopting a cost-benefit analysis method that quantifies the cost effectiveness of disaster prevention project. To demonstrate SIP-2, a case study of disaster risk reservoir maintenance projects completed in South Korea was adopted, through the following steps:

1) Among natural disaster risk reduction projects carried out by the South Korean government, information on disaster risk reservoir maintenance projects completed in 2009-2019 was collected.

2)    The loss rate of storm and flood insurance in the region where the flood damage occurred after the completion of the
maintenance project was investigated through KIDI.

3)    The amount of precipitation before and after the disaster risk reservoir maintenance project was investigated.
4)    Cost-benefit analysis was conducted to determine the economic feasibility of the maintenance project.

## 4 SIP-1: Improving the predictability of natural disaster-induced financial loss values using deep learning


SIP-1 aims to explore deep learning-driven modelling processes and develop an optimal learning model that can improve the
predictability of natural disaster-triggered financial losses. To demonstrate SIP-1, the loss amounts of storm and flood
insurance were learned, and the corresponding loss ratios were generalized associated with the selected risk indicators by the
property type. To scientifically validate the robustness of the learning model, the prediction results were compared with a
conventional parametric model underpinned by multiple regression analysis.

### 4.1 Data collection


A total of 458 storm and flood damage insurance claims for 11 years from 2009 to 2019 was collected from KIDI's data sets.
KIDI was established in 1983. It is an insurance professional service organization that develops insurance products, calculates
insurance rates, and protects the rights of policyholders. It also collects and manages various statistical data such as insurance
information and losses of each insurance company (Choi and Han, 2015). Storm and flood damage insurance, which reflects
the loss amount, is an insurance that compensates for property damage caused by natural disasters (e.g., typhoons, floods,
heavy rains, tsunamis, strong winds, storms, heavy snow, earthquakes, and so on). It has been implemented since 2006 under
the initiative of state and local governments (Kwon and Oh, 2018). The insurance payout amount is determined by objective
analysis of certified loss assessment service according to standardized procedures for each insurance company. Its reliability
is high (Kim et al., 2020). The collected data information includes the total loss amounts, the total net premiums, building
types, and location profiles, which is publicly available. The prediction model was trained, tested, and validated using losses
and natural disaster risk indicators.

The cost of loss due to natural disasters was divided by the total net premiums to calculate the ratio and then log-transformed,
which distribution of the data is shown in Figure 2. In addition, natural disaster risk indicators affecting insurance loss due to
natural disasters were collected. For natural disaster risk indicators, building type, wind speed, total rainfall, and peak ground
acceleration were selected as variables through past literature studies (Kim et al., 2017, 2019; Kim et al., 2020; Kim et al.,
2021). Figure 3, 4, 5, and 6 shows the distributions of the selected indicators. A description of variables is presented in Table
1. Building types were set as dummy variables that consist of residential buildings and greenhouses. Wind speed and the
maximum value of rainfalls were collected from the Korea Meteorological Administration (KMA). Peak ground accelerations

 were collected from the National Oceanic and Atmospheric Administration (NOAA). Accordingly, Table 2 summarises the

 descriptive statistics of variables.

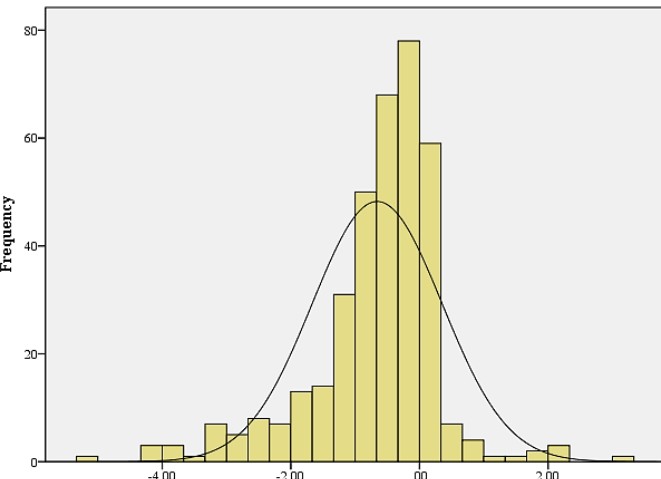

 Figure 2. Distribution of the insurance loss ratio record

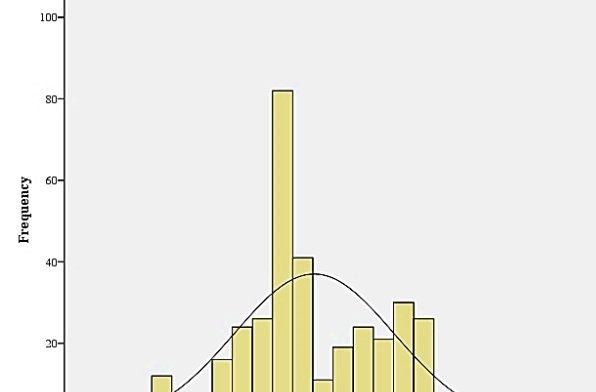

Figure 3. Distributions of the indicators to learn the loss ratios of Wind speed (m/s)

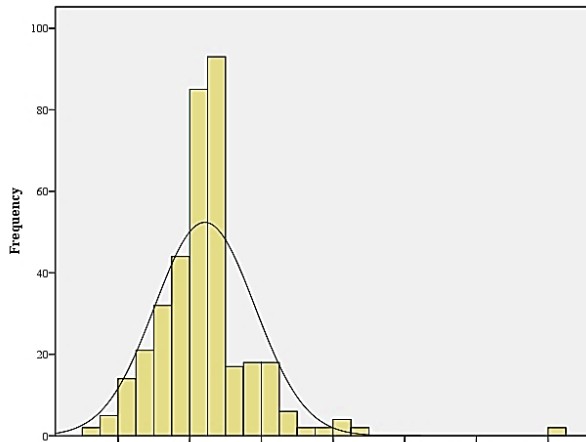

Figure 4. Distributions of the indicators to learn the loss ratios of Rainfall (mm/day)

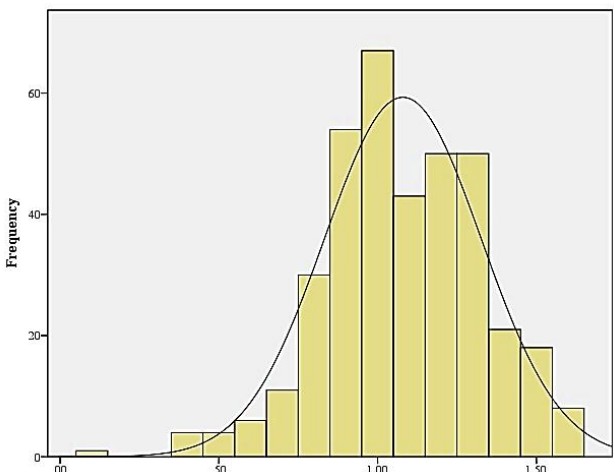 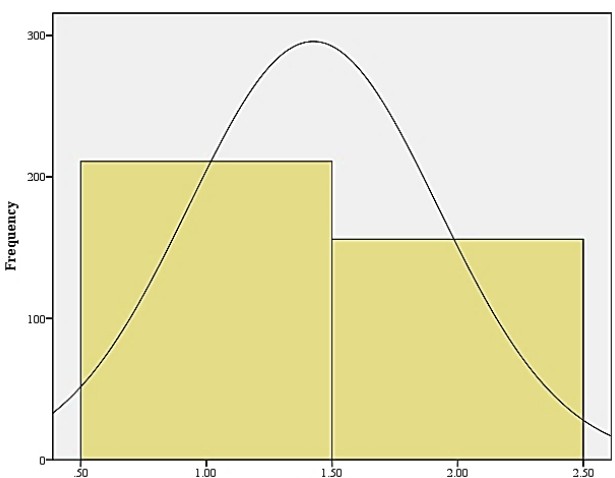

Figure 5. Distributions of the indicators to learn the loss ratios of Peak ground acceleration (g)

Figure 6. Distributions of the indicators to learn the loss ratios of Building type (1: residential, 2: greenhouse)



Table 1. Description of variables

| Variable | Explanation |
|---|---|
| Loss ratio | Total loss divided by the total net premium (Amount unit: KRW) |
| Building type | Buildings covered by storm and flood insurance (Categorical variable - Residential building: 1; Greenhouse: 2) |
| Wind speed | 10-minute average maximum wind speed (m/s) |
| Rainfall | Maximum precipitation per day (mm/day) |
| Peak ground acceleration | Value of peak ground acceleration (PGA) (g) |



Table 2. Descriptive statistics of variables by the building type (i.e., residential building and greenhouse)

| Variable (Unit) | Sample size | Minimum | Maximum | Mean | Std. Deviation |
|---|---|---|---|---|---|
| Loss ratio (Log-transformed value) | 458 | -5.12 | 3.17 | -0.66 | 1.01 |
| Wind speed (m/s) | 458 | 20.80 | 39.20 | 29.21 | 3.17 |
| Rainfall (mm/day) | 458 | 172.00 | 801.20 | 319.02 | 68.57 |
| Peak ground acceleration (g) | 458 | 0.10 | 1.60 | 1.10 | 0.25 |


## 4.2 Modeling deep neural networks

A deep learning algorithm is a neural network with many layers and various structures in general. Its use in research and industry for prediction and recognition has spread rapidly, proving its effectiveness (Kim et al., 2021). Deep learning algorithms are also widely used for regression analysis and type classification as a machine learning technique (Ajayi et al., 2019). Deep learning models have the same training framework as other types of neural networks. However, they can train large data sets more effectively with multiple hidden layers (Bae et al., 2021). Deep learning algorithms can be divided into deep neural network (DNN), generative adversarial network (GAN), recurrent neural network (RNN), convolutional neural network (CNN), and auto encoder (AE) according to their structure and processing method (Kim et al., 2021). Especially, DNN is used for cataloguing and prediction in various engineering and academic fields (Krizhevsky et al., 2012; Toya and Skidmore, 2007). Moreover, DNNs can be applied to train and model complex nonlinear relationships due to their multi-layered structures. Thus, in this study, a DNN model was accepted considering nonlinearity of collected loss data.

The learning performance of the model was appraised by measuring the values of root mean squared error (RMSE) and mean absolute error (MAE). RMSE and MAE are representative indicators of the size of the error by comparing the predicted result of an artificial neural network with the actual value (Daniell et al., 2011). RMSE is a value that measures the average error magnitude. MAE is a value obtained by converting the difference between the actual value and the predicted value into an absolute value and averaging it. Both indicators can be used to indicate that the prediction error decreases as the error value gets smaller (e.g., closer to zero).

The collected loss data were pre-processed using a *z*-score normalization method to adjust the unit and quantity of the data. The pre-processed completed input data were divided into a training set, a verification set, and a test set of data. The training set of data were used for learning of the DNN algorithm. The verification set of data were used to judge whether training was optimal and the test set of data were used to verify whether the developed model was finally trained for the purpose. In this study, considering the amount of data, 70% of the total data were set as training set of data and 30% of them were used as test set of data. Then 30% of training data were utilized as verification data.

The DNN model selected the optimal combination through a trial-and-error method since the DNN model could update the weights of neural network nodes with a backpropagation algorithm. Since various combinations were possible depending on the input variable and the output variable, it was necessary to find the optimal combination through the trial-and-error method. For such an optimal combination, it is necessary to define the network structure scenario for setting the number of layers and nodes and defining hyper parameters such as optimizers, activation functions, and dropouts (Cavallo et al., 2013). This study adopted a network structure scenario with three hidden layers considering data characteristics. Dropout is a regularization penalty to avoid overfitting. It was set to reduce prediction errors caused by overfitting. In this study, making an allowance for

the amount of training data, dropout was set to 0 and 0.2 and simulated. The ReLu (Rectified Linear Unit) function was utilized
as the activation function, a method of adjusting the weight of each node for optimal learning. The ReLu function allows the
input value to change when the input value is greater than 0 or less than 0. It was established to resolve the problem of gradient
loss of the existing Sigmoid function (Krizhevsky et al., 2012). The Adaptive Moment Estimation (Adam) method as accepted
as the optimizer (Krizhevsky et al., 2012). Optimizer is used for speed and stability of learning. The Adam Method is a widely
assumed algorithm since its development in 2015 (Kingma and Ba, 2015). The batch was defined as 5 as a data group
designation for efficient learning and the number of epochs was designated as 1,000 for the number of learning (Bae and Yoo,
2018; Ryu et al., 2018).

**4.3 Exploring DNNs and developing the DNN model**
Table 3 shows MAE and RMSE values according to the network structure and dropout. Amongst outcomes, the model with
the minimum MAE and RMSE was adopted as the final structure. As the number of hidden layer nodes increased, the MAE
and RMSE values fluctuated slightly. However, the number of hidden layer nodes was minimized at 25-25-25. When the
dropout was 0, MAE and RMSE values were commonly lesser than when the dropout was 0.2. It could be realized that when
the number of hidden layer nodes was 25-25-25 and the dropout was 0.0, both MAE and RMSE had minimum values.
Consequently, in the final structure, the number of nodes was 25-25-25 and the dropout was 0. Table 4 and Figure 7
demonstrate the network structure and hyper parameter configuration of the optimization model.

Table 3. Training results

| Network Structure Scenario | Dropout (0) | | Dropout (0.2) | |
|---|---|---|---|---|
| | MAE | RMSE | MAE | RMSE |
| 5-5-5 | 0.521 | 0.484 | 0.521 | 0.484 |
| 10-10-10 | 0.498 | 0.468 | 0.524 | 0.484 |
| 15-15-15 | 0.521 | 0.484 | 0.523 | 0.487 |
| 20-20-20 | 0.522 | 0.484 | 0.521 | 0.484 |
| 25-25-25 | 0.476 | 0.461 | 0.521 | 0.484 |
| 30-30-30 | 0.521 | 0.484 | 0.521 | 0.484 |
| 35-35-35 | 0.521 | 0.484 | 0.522 | 0.484 |
| 40-40-40 | 0.521 | 0.484 | 0.521 | 0.484 |
| 50-50-50 | 0.521 | 0.484 | 0.522 | 0.484 |


Table 4. Network structure and hyper parameter formation of the final model

| Category | Configuration | Feature |
|---|---|---|
| Network structure | Number of Hidden Layer | 3 |
| | Node | 25-25-25 |
| Hyper-parameter | Dropout | 0.0 |
| | Activation Function | ReLu (Rectified Linear Unit) |
| | Optimizer | Adam (Adaptive Moment Estimation) |
| | Epoch | 1000 |
| | Batch Size | 5 |


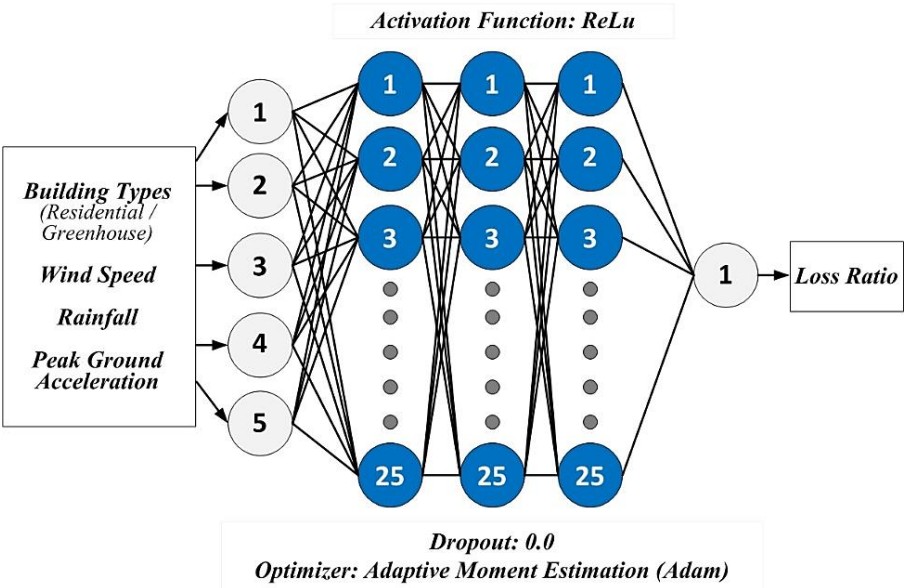

Figure 7. Final model of deep neural networks

**4.4 The robustness validation of the final DNN model**
An MRA (Multiple Regression Analysis) model was added for systematic validation of the final DNN model. MAE and RMSE
values of these two models were compared. The MRA method is widely adopted as an essential method for numerical
prediction models (Kim et al., 2021). Table 5 displays validation results of these models. Results of the DNN model showed
MAE of 0.531 and RMSE of 0.480 with the verification set of data. For the test set of data, results showed MAE of 0.452 and
RMSE of 0.435. There was no significant difference in MAE or RMSE between results with the test set of data and those with
the verification set of data since the overfitting problem of the final model could be overlooked. In addition, the MRA model

showed an MAE of 0.533 and a RMSE of 0.484. Equating outcomes of the DNN model and the MRA model, it was found that the DNN model had meaningfully minor prediction error rates of 15.2% MAE and 10.12% RMSE than the MRA model.

Table 5. Results with the validation set and test set of data

| | Validation Set | | Test Set | |
| --- | --- | --- | --- | --- |
| | MAE | RMSE | MAE | RMSE |
| DNN | 0.531 | 0.480 | 0.452 | 0.435 |
| MRA | - | - | 0.533 | 0.484 |
| DNN/MRA (%) | | | -15.20 | -10.12 |

## 5 SIP-2: Quantifying the cost effectiveness of natural disaster risk reduction projects using cost-benefit analysis

Management of a disaster risk reservoir is a part of the disaster prevention project. According to the Special Act on the Disaster Risk Reduction Project and Relocation Measures, the purpose of disaster prevention measures necessary for improving the disaster risk area is for fundamental prevention and permanent recovery of disasters. The disaster prevention project was started in 1998 when the Disaster Response Division of the Ministry of Government Administration and Home Affairs discovered disaster-prone facilities and areas with risk of human casualties and provided government funds for the maintenance of natural disaster risk areas for systematic management and prompt resolution of disaster risk factors (Lee, 2017). Disaster prevention projects include natural disaster risk improvement districts, disaster risk reservoirs, steep slope collapse risk areas, small rivers, and rainwater storage facilities (Kim et al., 2019). Given the significance of disaster prevention projects, SIP-2 examines economic effects through cost-benefit analysis of natural disaster risk reduction projects to reduce losses from natural disasters. To demonstrate SIP-2, a cost-benefit analysis was conducted for the natural disaster reduction project by comparing losses from storm and flood insurance before and after the disaster risk reservoir maintenance project.

### 5.1 Data collection and investigation of historical record

Among natural disaster risk reduction projects carried out by the South Korean government, the data set of disaster risk reservoir maintenance projects completed in 2009-2019 was extracted from the Public Data Portal (data.go.kr) managed by the South Korean government to collect and provide public data created or acquired by public institutions in one place. The system was established in 2011 to provide public data in the form of file data, visualization, and open API (Application Programming Interface) (Closs et al., 2014). During the study period of 2009-2019, 474 reservoirs were designated as disaster risk reservoirs and 290 maintenance projects were initiated. Among them, a total of 12 areas were flooded before and after the completion of the disaster risk reservoir maintenance project. Table 6 shows the loss rate and maximum precipitation at the time of flooding before and after completion of the maintenance projects in these 12 areas. Data about the loss amounts from storm and flood insurance were obtained from KIDI. Precipitation data were collected from KMA and the maximum daily

precipitation at the time of the flooding was used. Insured loss was expressed as a rate of the incurred loss divided by the accrued premium. The loss rate before the maintenance project was 34.32% on average, while that after the maintenance project was completed was 5.9% on average, showing a sharp decrease of 82.8% on average.

Table 6. Comparison of loss rate and precipitation before and after maintenance projects in flooded regions in South Korea

| No | Region | Loss rate | | Precipitation (mm/day) | |
|---|---|---|---|---|---|
| | | Before (%) | After (%) | Before | After |
| 1 | Yongin City | 47.40 | 20.60 | 425 | 188 |
| 2 | Nonsan City | 30.10 | 0.80 | 334 | 306 |
| 3 | Wanju-gun | 40.70 | 3.40 | 364 | 142 |
| 4 | Gangjin-gun | 76.30 | 0.40 | 235 | 166 |
| 5 | Sejong City | 7.30 | 4.90 | 257 | 223 |
| 6 | Muan-gun | 25.80 | 2.00 | 285 | 192 |
| 7 | Hampyeong-gun | 23.80 | 10.30 | 301 | 230 |
| 8 | Gyeongju City | 33.10 | 1.20 | 488 | 280 |
| 9 | Changwon City | 10.60 | 10.70 | 300 | 266 |
| 10 | Namhae City | 22.10 | 8.50 | 324 | 231 |
| 11 | Naju City | 53.90 | 5.10 | 330 | 106 |
| 12 | Goheung-gun | 40.70 | 3.00 | 325 | 249 |
| | Average (%) | 34.32 | 5.9 | 331 | 215 |
| | After/Before (%) | | 82.8 | | 35.0 |

**5.2 Cost-benefit analysis and results of natural disaster risk reduction projects**

As seen in Table 6, when data of precipitation as the main cause of flooding accidents during flood damage were compared, the average precipitation was 331 mm/day before the maintenance project and 215 mm/day after the maintenance project. It could be seen that the amount of precipitation was decreased by 35% when flood damage occurred after the maintenance project. The sharp decrease in the loss rate after the maintenance project could be due to not only the effect of maintenance project, but also decreased rainfalls. In turn, it is difficult to conclude that the decreased loss rate is due to the effect of reducing storm and flood damage caused by the maintenance project.

To analyze the cost effectiveness of the maintenance projects in flood regions, a cost-benefit analysis method using an equal-payment-series present-worth factor was adopted. The present-worth factor, assuming an annual loss rate i, is a coefficient used to find the present value corresponding to annual equivalent loss A for the next n years. Eq. (1) presents a widely used concept in economic analysis (Park & Sharp, 2021):

$$P = \frac{A[(1+i)^n - 1]}{i(1+i)^n}$$
(1)


Where:
*P*: Present value
*A*: Annual loss amount
*i*: Loss rate
*n*: Year

The initial cost of each maintenance project was collected through The Public Data Portal and the average cost of the
maintenance project was calculated. For the loss rate, the average loss rate of the loss area was used. For the annual loss amount,
the average annual loss for the study period (2009-2019) was used as seen in Table 7. However, it was assumed that no
additional costs incurred due to the maintenance project. Figure 8 shows calculation results before and after the maintenance
projects, which reveals that the loss amount becomes smaller after 8 years due to investment through the maintenance projects.

Table 7. Summary of inputs

| Input | Before | After |
|---|---|---|
| Initial cost | - | 22.088[*] |
| Loss rate | 0.343 | 0.059 |
| Annual loss amount | 0.371[*] | 0.006[*] |

[*] Billion KRW

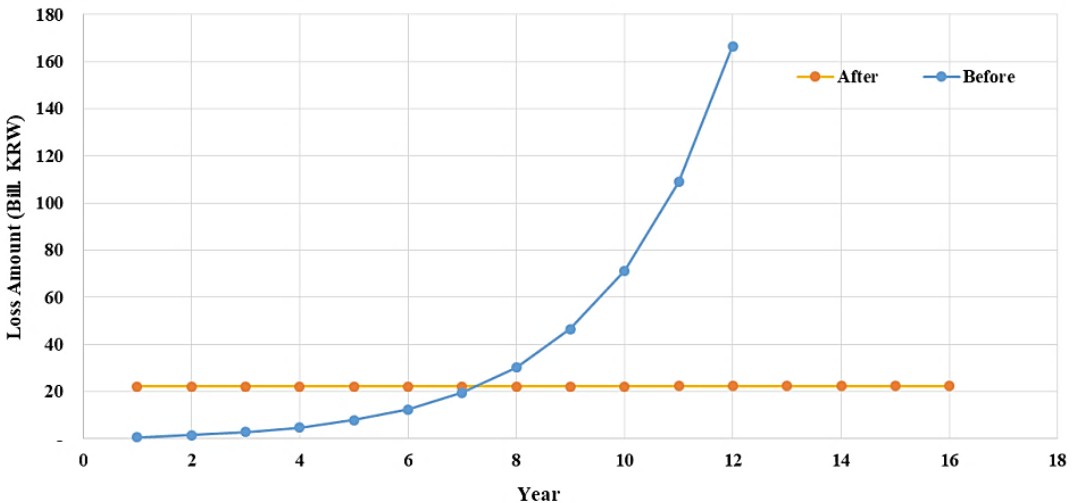

Figure 8. Comparison of losses before and after the maintenance projects

## 6 Discussion

Within the proposed strategic framework, SIP-1 developed an improved model for predicting economic losses due to natural disasters using the DNN algorithm. For model development, insurance company's storm and flood damage insurance loss records were used to collect economic losses caused by actual natural disasters. After developing a DNN model and training it with collected data, the final network model was selected by comparing with other DNN alternatives. To scientifically validate the improved predictability, the performance (i.e., actual-to-predicted comparison using MAE and RMSE methods) of the developed DNN model was compared with a parametric model underpinned by MRA. The results revealed that the DNN model was 15.2% less in the MAE and 10.12% less in the RMSE, compared to the MRA model. These results confirm that deep learning can produce more accurate and reliable prediction results of natural disaster-induced economic loss values associated with non-linear characteristics of risk indicators. It is noteworthy that the proposed implementation process is applicable to various natural disaster-triggered loss predictions, as the amount and its fluctuation of losses are diverse dependant on various types and strengths of natural disasters. In this sense, the proposed SIP-1 will help natural disaster risk managers predict the financial loss cost of natural disasters or develop an optimally customized prediction model by adopting deep learning. It can also be used as a reference when developing risk reduction investment plans or financial guideline in public and private sectors. For example, by applying this implementation process, it would be possible to estimate reliably the negative impact of natural disaster events on existing financial management practices and thus make decisions proactively on the most feasible risk reduction investment plan that can strengthen natural disaster risk management and reduce the amount of risk, ultimately reducing the economic loss caused by natural disasters. Based on the well-developed financial guideline, it would be possible to avoid any transfers of unexpected financial losses from insurance coverages or special purchases suitable for expected losses. Despite the merit of SIP-1, there still remain some limitations. First, owing to the limited data set, it was problematic to accumulate different data sets. Additional research in the future is needed to parallel and prove loss records in other countries or regions. In addition, further research is required to increase the amount of available data and upgrade the model through the introduction of additional variables to more precisely predict losses from natural disasters using deep learning algorithms.

Compared to SIP-1, SIP-2 proposed a new methodology that can quantify the cost effectiveness of natural disaster risk reduction projects through the cost-benefit analysis. To demonstrate SIP-2, among natural disaster risk reduction projects were implemented in South Korea, specific information of the disaster risk reservoir maintenance projects where flood damage occurred before and after completion was collected. Then, to identify benefits and costs, corresponding loss rates and daily precipitation amounts were investigated and compared at the project level. Lastly, the cost effectiveness of the projects was analyzed using a cost-benefit analysis method. As the result of cost-benefit analysis, in the short term, the loss after the maintenance project was greater than that before the maintenance project. However, this was reversed from 8 years after the maintenance project and the loss amount before the maintenance project was larger than that after the maintenance project.

Although it is difficult to expect profits from the maintenance project in the short term, it can be seen that the maintenance
project is economically beneficial in the long term (8 years or more). SIP-2 would be useful for making sounder decisions on
natural disaster management policy and natural disaster risk reduction project investment plans. Evaluating the effectiveness
of risk reduction through SIP-2 will lead to drastic investment, which will ultimately reduce the amount of natural disaster
risks. However, it should be noted that the study period shown in the SIP-2 case study was relatively short, while the location
of project samples was limited to South Korea. In addition, it was assumed that the inflation rate is identical during the study
period. In turn, it is necessary to conduct additional analyses considering various locations venerable to natural disasters in
other countries and more realistic financial loss values using a net present value concept.
**7 Conclusion**
Due to increasing threats to the life of general public and built assets from natural disasters, a variety of risk mitigation activities
are being carried out extensively. Given the continuous trend toward natural disaster risk mitigation, the significance of relevant
economic analyses has been underlined, against the limited public budget and its economic feasibility. To overcome this
difficulty, this study proposed a strategic framework for natural disaster risk mitigation, highlighting two different SIPs. SIP-
1 introduced more powerful method that can improve the predictability of natural disaster-triggered financial loss values using
deep learning, while SIP-2 highlighted the risk mitigation strategy at the project level, adopting a cost-benefit analysis method.
In SIP-1, a DNN model for natural disaster loss prediction was developed, and the improved predictability was validated by
comparing with MRA. The developed model learned and generalized the loss amount of natural disaster risk indicator facilities
(building type, wind speed, total rainfall, and peak ground acceleration) and wind and flood insurance. By evaluating learning
performances of 18 different DNN alternatives using RMSE and MAE values as representative evaluation indicators of deep
learning algorithms, 25-25-25 hidden layers with dropouts of 0.0 structure was selected as the optimal learning model. The
robustness of the developed model was technically validated by comparing RMSE and MAE values of a conventional
parametric model using a multiple regression analysis. Validation results confirmed that the non-parametric DNN model was
powerful for predicting non-linear characteristics of losses caused by natural disasters. In SIP-2, The cost-benefit analysis was
conducted on the disaster risk reservoir maintenance project that occurred before and after the completion of the flood damage.
As the result, it was difficult to expect profits from the maintenance business in the short term. However, in the long term
(more than 8 years), it was found that the maintenance business was economically profitable. The proposed framework is
unique as it provides a combinational approach to mitigating cost risk impacts of natural disasters at both financial loss and
project levels. Main findings of this study could be used as a guideline for decision-making of natural disaster management
policies and investment in natural disaster risk reduction projects. This study is its first kind and supporting the current
knowledge framework. This study will help practitioners quantify the loss from various natural disasters, while allowing them
to evaluate the cost effectiveness of risk reduction projects through a holistic approach.

## Code and data availability.

The data presented in this research are available from the first or corresponding author upon reasonable request.

## Author contributions.

**J.-M.**: contributed to the conceptualization and supervision; methodology development; data curation; investigation; project administration; resources and visualization; and writing the original manuscript and reviewing the revised manuscript. **S.-G**: contributed to data curation; investigation and validation; and reviewing the manuscript. **H.**: contributed to investigation; improving the literature review; and reviewing the manuscript. **J.**: contributed to strengthening research methodology and strategic framework design; visualization and validation; and reviewing and editing the manuscript as the corresponding author.

## Competing interests.

The authors declare that they have no conflict of interests.

## Acknowledgement

This research was supported by Research Funds of Mokpo National University in 2021.

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
