# Peer review of "Strategic Framework for Natural Disaster Risk Mitigation Using Deep Learning and Cost-Benefit Analysis"

_Natural Hazards and Earth System Sciences, 2021_

## Referee Comment (RC1)

[referee-annotated manuscript omitted]

---

## Author Comment (AC1)

Response to the **Reviewer #1**' comments concerning **NHESS-2021-294**

We sincerely thank the Reviewer 1 for the time in effort on reviewing our manuscript with many insightful comments. We believe that we have addressed each of the comments carefully and properly, and we will revise the manuscript, by fully reflecting those in the next stage of the revised manuscript submission. We hope that the changes listed below are acceptable for publication. Please note that the line numbers addressed in this letter indicates the numbers shown in the manuscript submitted initially as we are not allowed to upload the revised manuscript itself yet.

**General Comment:** … The paper deserves minor revisions as follows:

Thank you for positively viewing our research ideas, with minor revisions required. We appreciate very insightful comments listed below.

**Comment 1:** The title of the manuscript is too long, please consider shorter title such as: "Natural Disaster-Mitigation Using Deep Learning and Cost-Benefit Analysis".

Thank you for the suggestion. We have revised the title properly:

"*Strategic Framework for Natural Disaster Risk Mitigation Using Deep Learning and Cost-Benefit Analysis*"

**Comment 2:** The abstract does not reflect the novelty of the methodology. Please consider revision: stress the validity of findings and efficiency of the framework.

Thank you for this comment. To clarify the validity of findings and efficiency of the framework, the abstract has been re-written fully. In addition, to eliminate a logical fallacy of two aspects (i.e., stages 1 and 2) raised by the Reviewer 2, the abstract and the relevant main text body have been revised, especially the revised abstract reads:

"*Given trends in more frequent and severe natural disaster events, developing effective mitigation strategies is crucial to reduce negative economic impacts on built assets, due to the limited budget for rehabilitation. To address this need, this study aims to develop a strategic framework for natural disaster risk mitigation, highlighting two different strategic implementation processes (SIPs). SIP-1 is intended to improve the predictability of natural disaster-triggered financial loss model. To this end, SIP-1 develops a deep neural networks (DNN)-driven learning model that learns insurance loss amounts to generalize loss ratios, associated with major indicators including rainfall, wind, and ground acceleration. SIP-2 underlines the risk mitigation strategy at the project level, by adopting a cost-benefit analysis method that quantifies the cost effectiveness of disaster prevention projects. To demonstrate SIP-2, a case study of disaster risk reservoir projects in South Korea was adopted. The validated result of SIP-1 confirmed that the predictability of DNN is more accurate and reliable than traditional multiple regression analysis, while SIP-2 revealed that maintenance projects are economically more beneficial in the long-term as the loss amount becomes smaller after 8 years, coupled with the investment in the projects. The proposed framework is unique as it provides a combinational approach to mitigating cost risk impacts of natural disasters at both financial loss and project levels. This study is its first kind and will help practitioners quantify the loss from natural disasters, while allowing them to evaluate the cost effectiveness of risk reduction projects through a holistic approach.*"

**Comment 3:** The literature review may be extended to applications of Deep Learning to risk assessment – Please review and consider the following literature: (Khosravi et al. 2020; Zhang et al. 2022; Yi et al. 2020; Al Najar et al. 2021; Moishin et al. 2021; Shane Crawford et al. 2020; Sugiyarto and Rasjava 2020; Kim et al. 2021).

Thank you for the constructive suggestion. Accordingly, the statement of deep learning applications for risk assessment has been added in the section 1.2, by reviewing the recommended articles as follows:

[revised manuscript text omitted]

**Comment 4:** Please add a research framework diagram, emphasize the core phases of the methodology.

Thank you for this comment. To provide the intention of this study and the core phases of the methodology clearly, the research framework diagram has been added.

[Figure]

*Figure 1. Research framework*

As stated earlier by responding to comment 1, we have highlighted the main purpose of each of the SIPs.

Thank you.

**Comment 5:** The training phase of the DNN must be significantly improved: please provide figures with distributions of the core variables, please provide distribution figures of the training data (Loss ratio, building type, maximum wind speed, rainfall and PGA), the MRA model and the DNN output. This is essential to for legibility and scientific soundness.

Thank you for the comments. We have provided figures of distributions of learning output (i.e., loss ratio) and indicators, as follows:

[Figure]

*Figure 2. Distribution of insurance loss ratio record*

[Figure]

(a) Wind speed (m/s)

(b) Rainfall (mm/day)

(c) Peak ground acceleration (g)

(d) Building type (1: residential; 2: greenhouse)

*Figure 3. Distributions of indicators to learn the loss ratios*

On the other hand, when it comes to MRA-to-DNN output comparison, the main purpose of SIP 1 was to introduce more powerful method that can improve the predictability of natural disaster-triggered financial loss values, comparing with a traditional method like MRA, not focused on the achievement of predicted values themselves. In this sense, we have provided Table 5, which would be effective enough to show the comparison results of the predictability scientifically, with the error values.

Alternately, to provide more clear result of the final DNN, we have provided the network architecture graphically, along with Table 4.

[Figure]

*Figure 4. Final model of deep neural networks*

We hope that this is acceptable for your consideration.

**Comment 6:** Please add keyworks.

Thank you for pointing this out. We have addressed this comment, by adding keywords as follows:

*Keywords: Natural disaster; risk mitigation strategy; economic damage; deep learning; cost-benefit analysis*

**Comment 7:** Please see further remarks and corrections in the attached file.

Overall, thank you for your insight review. We would appreciate it. Accordingly, we have corrected minor errors as well.
- Line 81: The duplicated word, "earthquakes" has been eliminated.
- Line 82: The transition "In addition" has been replaced with "Furthermore."
- The terms between "cost-benefit" vs "benefit-cost": We have made the use of term consistently, with "cost-benefit" throughout the paper.
- Line 125: "disaster reduction" has been replaced with "disaster risk reduction."
- Line 278 – Line 279 corrected to "due to not only the effect of maintenance project, but also decreased rainfalls."
- Line 297: The typo has been corrected (i.e., table 7 → Table 7).
- Line 300 corrected to "projects."
- Line 350: "to reduce these threats" has been eliminated.
- Line 368: The typo has been corrected (i.e., an → can).

---

## Author Comment (AC2)

Response to the **Reviewer #2'** comments concerning **NHESS-2021-294**

We sincerely thank the Reviewer 2 for the time in effort on reviewing our manuscript with many insightful comments. We believe that we have addressed each of the comments carefully and properly, and we will revise the manuscript, by fully reflecting those in the next stage of the revised manuscript submission. We hope that the changes listed below are acceptable for publication. Please note that the line numbers addressed in this letter indicates the numbers shown in the manuscript submitted initially as we are not allowed to upload the revised manuscript itself yet.

**General Comment:** Gradual increases in the frequency and severity of natural disasters increase loss risk of human life and property. It is important to estimate the damages caused by the disaster before such events occurs. The authors attempted to develop a strategic evaluation framework and analyzed the natural disaster risk mitigation strategies by DNN methods with data of insured loss amounts between 2009 to 2019. The ideas is scientific and topic is suitable for NHESS.

Thank you for positively viewing our research ideas, with very insightful comments listed below. We appreciate it.

**Comment 1:** The quality of the data determines whether the analysis results are scientific. The authors carry out the research with data on claim payout for storm and flood damage insurance from the Korea Insurance Development Institute. But how many pieces? What information in the data? The author only gave the data name and resources. It is necessary to describe the data information in detail.

We do believe that certain KIDI data/information sets not used in this study are beyond scope of this study. Indeed, Table 1 describes the information of data, while Table 2 includes the sample size, 458 samples obtained from KIDK for this study.

Nevertheless, to address the comment, we have added explanation about the data information, which reads between Line 172 and Line 173:

"*The collected data information includes the total loss amounts, the total net premiums, building types, and location profiles, which is publicly available.*"

In addition, to provide the data information in detail, we have added new figures of distributions of training data sets and learning output (i.e., loss ratio), as follows:

[Figure]

*Figure 2. Distribution of insurance loss ratio record*

[Figure]

(a) Wind speed (m/s)

(b) Rainfall (mm/day)

(c) Peak ground acceleration (g)

(d) Building type (1: residential; 2: greenhouse)

*Figure 3. Distributions of indicators to learn the loss ratios*

**Comment 2:** The data collected are claim payout for storm and flood damage. But the selected disaster factors include wind speed and Peak Ground Acceleration. What is the reason? What is the relationship between them?

Indeed, in the manuscript, we have already addressed various cases of natural disasters that can be covered by insurance (Line 168 – Line 171), which reads "Storm and flood damage insurance, which reflects the loss amount, is an insurance that compensates for property damage caused by natural disasters (e.g., typhoons, floods, heavy rains, tsunamis, strong winds, storms, heavy snow, earthquakes, and so on). It has been implemented since 2006 under the initiative of state and local governments (Kwon and Oh, 2018)."

Given the information, we have selected "wind speeds", "rainfall" and *"peak ground acceleration as the indicator for the loss amount caused by earthquakes"* by affected building types. To support our justification, we have already cited proven previous studies (Line 178: Kim et al., 2017; Kim et al., 2020; Kim et al., 2021) in the main text body.

We hope that it clarifies.

**Comment 3:** It is an interesting attempt to use DNN to predict disaster loss. The authors just trained the model and compared it with MRA model. I think it is import to analyze it in detail and use it in resent disaster cases.

Indeed, the main purpose of developing the DNN model as the strategic implementation process (SIP 1) in the proposed framework is to introduce more powerful method that can improve the predictability of natural disaster-triggered financial loss values, comparing with a traditional method like MRA, not about its applicability of the model to other recent cases. The result of SIP 1 confirms that the predictability of DNN is more accurate and reliable than traditional multiple regression analysis.

We hope that this is acceptable and reasonable. Thank you.

**Comment 4:** The authors describe two stages in their work. Stage I is to create a predictive model based on a deep learning algorithm with data on claim payout for storm and flood damage insurance. Stage II is to analyzed the cost-benefit with data on natural disaster risk reduction projects. There seems to be no logical relationship between the two. What is the relationship between the two stages? The authors should describe it clearly.

Thank you for this comment. To provide the intention of this study and the core phases of the methodology clearly, we have replaced "stages (sequential process)" with "strategic implementation process (SIP)".

In other words, the revised SIPs are not sequentially related, but these represent the implementation processes in the proposed strategic framework. Accordingly, we have revised the abstract and the relevant main text body, while adding the research framework diagram.

[Figure]

*Figure 1. Research framework*

The Revised Abstract:

"*Given trends in more frequent and severe natural disaster events, developing effective mitigation strategies is crucial to reduce negative economic impacts on built assets, due to the limited budget for rehabilitation. To address this need, this study aims to develop a strategic framework for natural disaster risk mitigation, highlighting two different strategic implementation processes (SIPs). SIP-1 is intended to improve the predictability of natural disaster-triggered financial loss model. To this end, SIP-1 develops a deep neural networks (DNN)-driven learning model that learns insurance loss amounts to generalize loss ratios, associated with major indicators including rainfall, wind, and ground acceleration. SIP-2 underlines the risk mitigation strategy at the project level, by adopting a cost-benefit analysis method that quantifies the cost effectiveness of disaster prevention projects. To demonstrate SIP-2, a case study of disaster risk reservoir projects in South Korea was adopted. The validated result of SIP-1 confirmed that the predictability of DNN is more accurate and reliable than traditional multiple regression analysis, while SIP-2 revealed that maintenance projects are economically more beneficial in the long-term as the loss amount becomes smaller after 8 years, coupled with the investment in the projects. The proposed framework is unique as it provides a*

*combinational approach to mitigating cost risk impacts of natural disasters at both financial loss and project levels. This study is its first kind and will help practitioners quantify the loss from natural disasters, while allowing them to evaluate the cost effectiveness of risk reduction projects through a holistic approach.*"

**Comment 5:** Sections about the methods and results should be revised. The authors mixes methods and results together in Section 3 and Section 4, which are nor clearly presented.

Thank you for the comment. Indeed, the content structure of research has been highlighted by each SIP. In turn, we have addressed each SIP in each of the sections. Given the intentions, we do believe that the section 3 (SIP 1) has been structured clearly, from the data collection (section 3.1) to the model validation (section 3.4). Nevertheless, to clarify the main purpose of SIP 1, we have revised the corresponding section title to "*3 Improving the predictability of natural disaster-induced financial loss values using deep learning*".

we do agree with the reviewer's comment on the section 4. Hence, we have re-structured and re-written the section 4 to clearly present the SIP 2, as follows:

"*4 SIP 2: Quantifying the cost effectiveness of natural disaster risk reduction projects using cost-benefit analysis*

*Management of a disaster risk reservoir is a part of the disaster prevention project. According to the Special Act on the Disaster Risk Reduction Project and Relocation Measures, the purpose of disaster prevention measures necessary for improving the disaster risk area is for fundamental prevention and permanent recovery of disasters. The disaster prevention project was started in 1998 when the Disaster Response Division of the Ministry of Government Administration and Home Affairs discovered disaster-prone facilities and areas with risk of human casualties and provided government funds for the maintenance of natural disaster risk areas for systematic management and prompt resolution of disaster risk factors (Lee, 2017). Disaster prevention projects include natural disaster risk improvement districts, disaster risk reservoirs, steep slope collapse risk areas, small rivers, and rainwater storage facilities (Kim et al., 2019). Given the significance of disaster prevention projects, SIP 2 examines economic effects through cost-benefit analysis of natural disaster risk reduction projects to reduce losses from natural disasters. To demonstrate SIP 2, a cost-benefit analysis was conducted for the natural disaster reduction project by comparing losses from storm and flood insurance before and after the disaster risk reservoir maintenance project.*

*4.1 Data collection and investigation of historical record*
*To gather data, among natural disaster risk reduction projects carried out by the South Korean government, information on disaster risk reservoir maintenance projects completed in 2009-2019 was collected from the Public Data Portal (data.go.kr) managed by the South Korean government to collect and provide public data created or acquired by public institutions in one place. The system was established in 2011 to provide public data in the form of file data, visualization, and open API (Application Programming Interface) (Closs et al., 2014). During the study period of 2009-2019, 474 reservoirs were designated as disaster risk reservoirs and 290 maintenance projects were initiated. Among them, a total of 12 areas were flooded before and after the completion of the disaster risk reservoir maintenance project. Table 6 shows the loss rate and maximum precipitation at the time of flooding before and after completion of the maintenance projects in these 12 areas. Data about the loss amounts from storm and flood insurance were obtained from KIDI. Precipitation data were collected from KMA and the maximum daily precipitation at the time of the flooding was used. Insured loss was expressed as a rate of the incurred loss divided by the accrued premium. The loss rate before the maintenance project was 34.32% on average, while that after the maintenance project was completed was 5.9% on average, showing a sharp decrease of 82.8% on average.*

*<Table 6 here>*

*4.2 Cost-benefit analysis and results of natural disaster risk reduction projects*
*As seen in Table 6, when data of precipitation as the main cause of flooding accidents during flood damage were compared, the average precipitation was 331 mm/day before the maintenance project and 215 mm/day*

*after the maintenance project. It could be seen that the amount of precipitation was decreased by 35% when flood damage occurred after the maintenance project. The sharp decrease in the loss rate after the maintenance project could be due to not only the effect of maintenance project, but also decreased rainfalls. In turn, it is difficult to conclude that the decreased loss rate is due to the effect of reducing storm and flood damage caused by the maintenance project.*

*To analyze the cost effectiveness of the maintenance projects in flood regions, a cost-benefit analysis method using an equal-payment-series present-worth factor was adopted. The present-worth factor, assuming an annual loss rate i, is a coefficient used to find the present value corresponding to annual equivalent loss A for the next n years. Eq. (1) presents a widely used concept in economic analysis (Park & Sharp, 2021):*

*<Equation (1) here>*

*The initial cost of each maintenance project was collected through The Public Data Portal and the average cost of the maintenance project was calculated. For the loss rate, the average loss rate of the loss area was used. For the annual loss amount, the average annual loss for the study period (2009-2019) was used. Figure 1 (now "Figure 5" in the revised manuscript) shows calculation results before and after the maintenance project. As can be seen from Figure 1, the loss amount becomes smaller after 8 years due to investment through the maintenance project…."*

---

## Author Response (AR1)

**Response to the **Reviewer #1**' comments concerning **NHESS-2021-294**

We sincerely thank the Reviewer 2 for the time in effort on reviewing our manuscript with many insightful comments. We believe that we have addressed each of the comments carefully and properly, while improving the quality of paper significantly. We hope that the changes listed below are acceptable for publication.

In addition, we have made significant changes in all the relevant main text body, which could be aligned well with our responses to the comments. All the changes made were highlighted using "Track Changes" mode of Microsoft Word in the revised manuscript attached.

General Comment: ... The paper deserves minor revisions as follows:

Thank you for positively viewing our research ideas, with minor revisions required. We appreciate very insightful comments listed below.

**Comment 1:** The title of the manuscript is too long, please consider shorter title such as: "Natural Disaster-Mitigation Using Deep Learning and Cost-Benefit Analysis".

Thank you for the suggestion. We have revised the title properly:

"Strategic Framework for Natural Disaster Risk Mitigation Using Deep Learning and Cost-Benefit Analysis"

**Comment 2:** The abstract does not reflect the novelty of the methodology. Please consider revision: stress the validity of findings and efficiency of the framework.

Thank you for this comment. To clarify the validity of findings and efficiency of the framework, the abstract has been re-written fully. In addition, to eliminate a logical fallacy of two aspects (i.e., stages 1 and 2) raised by the Reviewer 2, the abstract and the relevant main text body have been revised. Especially the revised abstract reads:

"Given trends in more frequent and severe natural disaster events, developing effective risk mitigation strategies is crucial to reduce negative economic impacts, due to the limited budget for rehabilitation. To address this need, this study aims to develop a strategic framework for natural disaster risk mitigation, highlighting two different strategic implementation processes (SIPs). SIP-1 is intended to improve the predictability of natural disaster-triggered financial losses using deep learning. To demonstrate SIP-1, SIP-1 explores deep neural networks (DNNs) that learn storm and flood insurance loss ratios associated with selected major indicators and then develops an optimal DNN model. SIP-2 underlines the risk mitigation strategy at the project level, by adopting a cost-benefit analysis method that quantifies the cost effectiveness of disaster prevention projects. In SIP-2, a case study of disaster risk reservoir projects in South Korea was adopted. The validated result of SIP-1 confirmed that the predictability of the developed DNN is more accurate and reliable than a traditional parametric model, while SIP-2 revealed that maintenance projects are economically more beneficial in the long-term as the loss amount becomes smaller after 8 years, coupled with the investment in the projects. The proposed framework is unique as it provides a combinational approach to mitigating economic damages caused by natural disasters at both financial loss and project levels. This study is its first kind and will help practitioners quantify the loss from natural disasters, while allowing them to evaluate the cost effectiveness of risk reduction projects through a holistic approach."

**Comment 3:** The literature review may be extended to applications of Deep Learning to risk assessment – Please review and consider the following literature: (Khosravi et al. 2020; Zhang et al. 2022; Yi et al. 2020; Al Najar et al. 2021; Moishin et al. 2021; Shane Crawford et al. 2020; Sugiyarto and Rasjava 2020; Kim et al. 2021).

Thank you for the constructive suggestion. Accordingly, the statement of deep learning applications for risk assessment has been added in the section 1.2, by reviewing the recommended articles as follows:

- Al Najar, M., Thoumyre, G., Bergsma, E. W., Almar, R., Benshila, R., and Wilson, D. G.: Satellite derived bathymetry using deep learning. Machine Learning, 1-24, 2021.
- Khosravi, K., Panahi, M., Golkarian, A., Keesstra, S. D., Saco, P. M., Bui, D. T., and Lee, S.: Convolutional neural network approach for spatial prediction of flood hazard at national scale of Iran. Journal of Hydrology, 591, 125552, 2020.
- Kim, J., Yum, S., Son, S., Son, K., and Bae, J.: Modeling deep neural networks to learn maintenance and repair costs of educational facilities. Buildings, 11(4), 165, 2021.
- Moishin, M., Deo, R. C., Prasad, R., Raj, N., and Abdulla, S.: Designing deep-based learning flood forecast model with ConvLSTM hybrid algorithm. IEEE Access, 9, 50982-50993, 2021.
- Rasjava, A. R. I., Sugiyarto, A. W., Kurniasari, Y., and Ramadhan, S. Y.: Detection of Rice Plants Diseases Using Convolutional Neural Network (CNN). In Proceeding International Conference on Science and Engineering, 3, 393-396, 2020.
- Shane Crawford, P., Hainen, A. M., Graettinger, A. J., van de Lindt, J. W., and Powell, L.: Discrete-outcome analysis of tornado damage following the 2011 Tuscaloosa, Alabama, tornado. Natural Hazards Review, 21(4), 04020040, 2020.
- Yi, Y. and Zhang, W.: A new deep-learning-based approach for earthquake-triggered landslide detection from singletemporal RapidEye satellite imagery. IEEE Journal of Selected Topics in Applied Earth Observations and Remote Sensing, 13, 6166-6176, 2020.
- Zhang, Y., Shi, X., Zhang, H., Cao, Y., and Terzija, V.: Review on deep learning applications in frequency analysis and control of modern power system. International Journal of Electrical Power & Energy Systems, 136, 107744, 2022.

Accordingly, the extended review of literature reads (Line 87 – Line 115 under the "No Markup" mode in the revised manuscript):

"Given the increasing demand, many research efforts on applying deep learning techniques for risk assessment were made recently (Al Najar et al. 2021; Khosravi et al. 2020; Kim et al. 2021; Moishin et al. 2021; Shane Crawford et al. 2020; Sugiyarto and Rasjava 2020; Yi et al. 2020; Zhang et al. 2022). Especially, for improved natural disaster risk assessment and mitigation, neural networks have been widely used for deep learning in various ways (Khosravi et al. 2020; Moishin et al. 2021; Shane Crawford et al. 2020; Yi et al. 2020; Moishin et al. 2021; Shane Crawford et al. 2020; Yi et al. 2020; Noishin et al. 2021; Shane Crawford et al. 2020; Yi et al. 2020; Yi et al. 2020).

Some researchers developed deep learning models to predict flood events (Khosravi et al. 2020; Moishin et al. 2021). Khosravi et al. (2020) developed a flood susceptibility map using convolutional neural networks (CNN). More specifically, 769 historical flood locations in Iran were trained and tested based on amounts of soil moisture, slopes, curvatures, altitudes, rainfalls, geology, land use and vegetation, distances from roads and rivers. In addition, a hybrid deep learning algorithm integrating the merits of CNN and long short-term memory (LSTM) networks was built to manage flood risks by predicting future flood events, by training and testing daily rainfall data obtained from 11 sites in Fiji between 1990 and 2019 (Moishin et al. 2021).

Other previous studies focused on post-disaster detection caused by landslides or tornados, which uses remote sensed data collected from satellites for deep learning (Al Najar et al. 2021; Shane Crawford et al. 2020; Yi et al. 2020). Shane Crawford et al. (2020) adopted CNN to classify damages of 15,945 buildings affected by the 2011 Tuscaloosa tornado in Alabama. To this end, the authors used satellited-driven images of trees as the damage classification indicator to estimate wind speeds. In addition, satellite images were embraced into the CNN-driven deep learning process to detect earthquake-induced landslides in China (Yi et al. 2020). More recently, Al Najar et al. (2021) estimated accurately ocean depths simulating remote sensed images using a deep learning technique, which overcomes drawbacks of traditional bathymetry measurement activities to track the physical evolution of coastal areas against any potential natural disasters

or extreme storm events. Previous studies reviewed reveal consistently that deep learning techniques can overcome shortcomings of existing methods and thus to provide more accurate and reliable decision-support models for risk assessment and risk-informed mitigation strategies.

In addition to applications of deep learning for location detection or event prediction-focused, as stated earlier, it is important to quantify negative economic impacts caused by natural disasters. Given the importance of economic damage aspects, Kim et al. (2021) applied a deep learning technique as a costeffective and risk-informed facilities management solution. In detail, the authors generalized maintenance and repair costs of educational facilities in Canada, using deep neural networks that learn sets of maintenance and repair records, asset values, natural hazards such as tornados, lightening, hails, floods, and storms. In this sense, this study proposed a deep learning modeling framework to predict financial losses caused by natural disasters."

**Comment 4: Please add a research framework diagram, emphasize the core phases of the methodology.**

Thank you for this comment. To provide the intention of this study and the core phases of the methodology clearly, the research framework diagram has been added.

Figure 1. Research framework

With the updated research framework, all the relevant main text bodies have been improved. Due to the large volume of changes, we could not cut and paste them in this letter.

All the changes made were highlighted using "Track Changes" mode of Microsoft Word in the revised manuscript attached.

Thank you.

**Comment 5:** The training phase of the DNN must be significantly improved: please provide figures with distributions of the core variables, please provide distribution figures of the training data (Loss ratio, building type, maximum wind speed, rainfall and PGA), the MRA model and the DNN output. This is essential to for legibility and scientific soundness.

Thank you for the comments. We have provided figures of distributions of learning output (i.e., loss ratio) and indicators, as follows:

Figure 2. Distribution of insurance loss ratio record